materials science

full Heusler alloys, electronic band structure, uniform strain, tetragonal strain, phase transformation

**Author for correspondence:**
Rabah Khenata
e-mail: rabah_khenata@univ-mascara.dz

This article has been edited by the Royal Society of Chemistry, including the commissioning, peer review process and editorial aspects up to the point of acceptance.

# Investigation of the structural competing and atomic ordering in Heusler compounds Fe$_2$NiSi and Ni$_2$FeSi under strain condition

Tie Yang[1], Liyu Hao[1], Rabah Khenata[2] and Xiaotian Wang[1]

[1]School of Physical Science and Technology, Southwest University, Chongqing 400715, People's Republic of China
[2]Laboratoire de Physique Quantique de la Matiere et de Modelisation Mathematique, Universite de Mascara, Mascara 29000, Algeria

(iD) TY, 0000-0001-8124-7942; RK, 0000-0002-5573-1711

The structural competing and atomic ordering of the full Heusler compounds Fe$_2$NiSi and Ni$_2$FeSi under uniform and tetragonal strains have been systematically studied by the first-principles calculation. Both Fe$_2$NiSi and Ni$_2$FeSi have the XA structure in cubic phase and they show metallic band structures and large magnetic moments (greater than $3\mu_B$) at equilibrium condition. Tetragonal distortion can further decrease the total energy, leading to the possible phase transformation. Furthermore, different atom reordering behaviours have been observed: for Fe$_2$NiSi, atoms reorder from cubic XA-type to tetragonal L1$_0$-type; for Ni$_2$FeSi, there is only structural transformation without atom reordering. The total magnetic moments of Fe$_2$NiSi and Ni$_2$FeSi are mainly contributed by Fe atoms, and Si atom can strongly suppress the moments of Fe atoms when it is present in the nearest neighbours of Fe atoms. With the applied strain, the distance between Fe and Si atoms play an important role for the magnetic moment variation of Fe atom. Moreover, the metallic band nature is maintained for Fe$_2$NiSi and Ni$_2$FeSi under both uniform and tetragonal strains. This study provides a detailed theoretical analysis about the full Heusler compounds Fe$_2$NiSi and Ni$_2$FeSi under strain conditions.

## 1. Introduction

During last decades, Heusler alloys have received tremendous research interests and become of great importance for the

development of new functional materials due to their various and special properties, such as half-metallicity [1–8], semi-metallicity [9,10], thermoelectricity [11–14], spin-gapless semiconductivity [1,15–22], ferromagnetism [23–29] and topological insulativity [30–32]. Their applications spread into different fields, mainly including spintronics and magnetoelectronics [33–36]. Conventional Heusler materials comprise a large group of intermetallic compounds and they can be divided into two groups: half Heusler with stoichiometric compositions XYZ and full Heusler with $X_2YZ$, where X and Y are transition metal elements and Z is a main group element. A lot of experimental effort has been dedicated to develop novel Heusler alloys [37–40]. Especially with the rapid growth of microengineering and nanotechnology, different fabrication techniques and processes are widely available, like melt-spinning, arc-melting, magnetron sputtering, ball-milling and chemical coprecipitation. In the meantime, extensive theoretical calculations have also been devoted to study the properties of Heusler compounds and even design and predict new Heusler alloys [41–49].

The physical properties of Heusler compounds are directly related to their highly ordered structure. Typically, Heusler alloys crystallize in the face-centred cubic structure with two possible atomic orderings [27,50]: $Cu_2MnAl$-type and $Hg_2CuTi$-type. On the other hand, several studies have demonstrated that a large number of Heusler compounds have the tetragonal structure as the ground state [4,49,51–55]. Also, there could be various defects present in both structures, including disorder, antisite, swap and vacancy. Consequently, their properties would be strongly influenced by these structures, atom orderings and defects [37,54,56–61]. In particular, for the development of magnetic-tunnel-junction memory material and ferromagnetic shape memory material, tetragonal structure Heusler with large magnetization is preferable. The group of Fe-based and Ni-based Heusler compounds are good candidates for this purpose, such as $Fe_2CrGa$ has different atomic configuration dependent on the preparation methods, and its magnetization can be significantly enhanced by either Fe-Ga or Cr-Ga disorder [60]; $Fe_2MnGa$ has martensitic transformation induced by magnetic field, and it is accompanied by spontaneous magnetization [62]; $Fe_2CrAl$ prepared by ball-milling method has partially disordered $B2_{CD}$ structure, and it shows both higher Curie temperature and magnetic moments [59]; $Ni_2MnGa$ undergoes a phase transition from cubic structure at high temperature to tetragonal structure at low temperature [63]; $Mn_2NiGa$ has the stable tetragonal phase compared to the cubic phase, and it is ferromagnetic in both phases with different magnetic moment [64]. In combination of Fe and Ni together, several experimental studies have synthesized $Fe_2NiZ$ (Z = Al, Ga, Si and Ge) [37,65,66] along with some theoretical investigations on their electronic and magnetic properties under different structures [67,68]. Results show antisite disorder in $Fe_2NiGe$ and $Fe_2NiGa$ tends to enhance the stability of cubic structure [69,70], which is contradictory to the minimum total energy configuration in tetragonal phase when being chemically ordered. In particular, $Fe_2NiSi$ shows different structures under different preparation processes, indicating the complicated atomic ordering in this compound. For the purpose to better understand the different structural configuration and its impact on the electronic and magnetic properties of full Heusler compounds $Fe_2NiSi$ and $Ni_2FeSi$, we employ first-principles calculation. Two different structures of cubic phase and tetragonal phase with two different atomic orderings of $Cu_2MnAl$-type and $Hg_2CuTi$-type have been investigated. Furthermore, uniform and tetragonal strain conditions have also been considered and discussed.

# 2. Computational details

By using the pseudo-potential plane-wave method based on density functional theory, we have performed the first-principles calculation [71] with the Cambridge Serial Total Energy Package [72] to study the structural, electronic and magnetic properties of the full Heusler compounds $Fe_2NiSi$ and $Ni_2FeSi$. The Perdew–Burke–Ernzerhof functional within the generalized gradient approximation (GGA) [73] and the ultrasoft pseudo potential [74] are selected to describe the exchange-correlation potential and the interaction between the atomic core and the valence electrons. After initial convergence test, a plane-wave cut-off energy of 500 eV and a specific $k$-point mesh using a $15 \times 15 \times 15$ Monkhorst-Pack grid were applied for all calculations. The self-consistent field tolerance was set as a total energy difference smaller than $1 \times 10^{-6}$ eV $atom^{-1}$.

# 3. Results and discussion

## 3.1. Structure competing and equilibrium lattice

The full Heusler compound is normally presented by a generic formula as $X_2YZ$ and has a cubic structure with four interpenetrating face-centred cubic sublattices, which can be defined by the Wyckoff

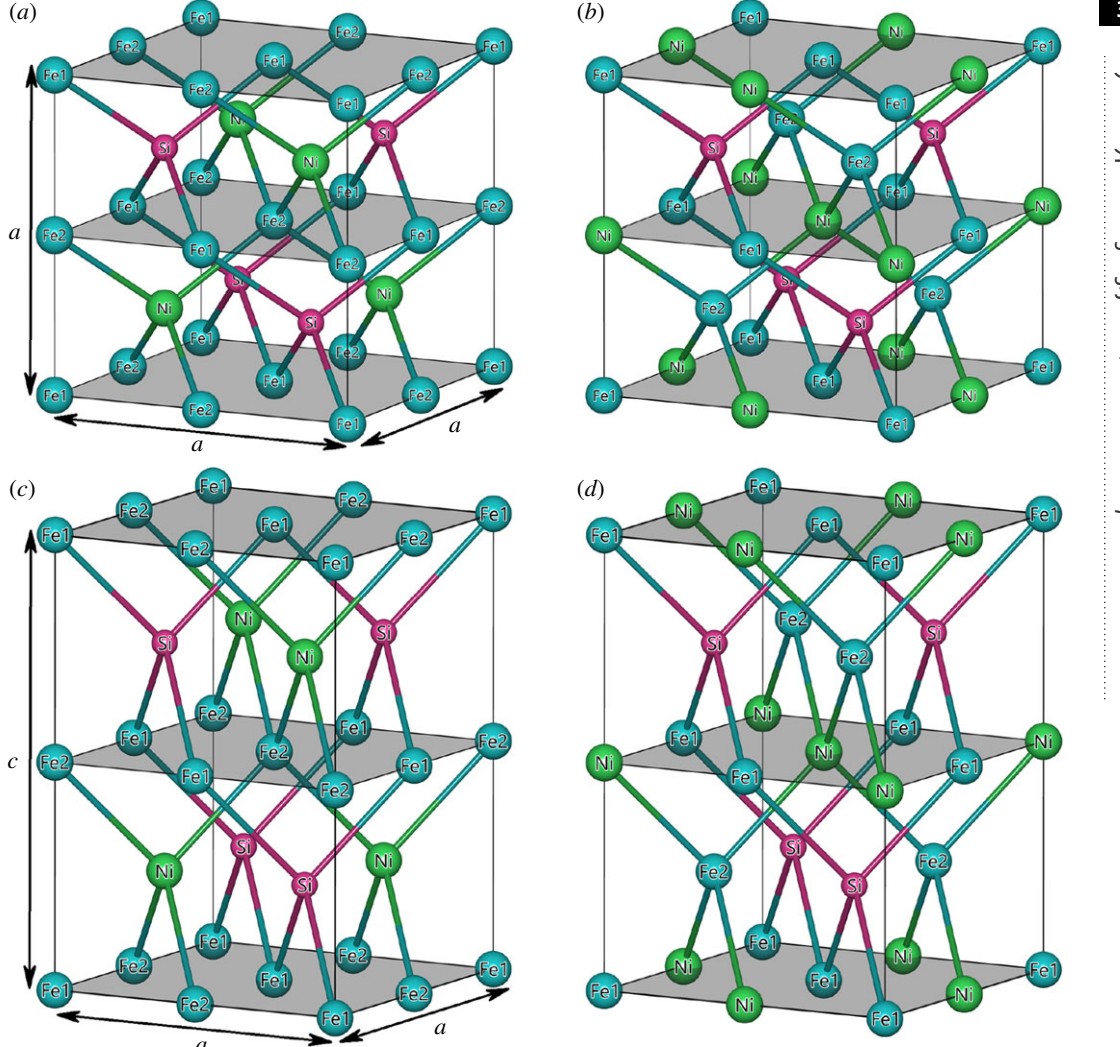

**Figure 1.** Four different crystal structures of full Heusler alloy Fe$_2$NiSi: (*a*) cubic L2$_1$-type, (*b*) cubic XA-type, (*c*) tetragonal L1$_0$-type and (*d*) tetragonal XA-type.

coordinates as A(0, 0, 0), B(0.25, 0.25, 0.25), C(0.5, 0.5, 0.5) and D(0.75, 0.75, 0.75). X and Y are transition metal elements and they enter A, B and C sites; Z is from main group elements and it enters D site. Different arrangements of transition metal elements X and Y at A, B and C Wyckoff positions generally result into two different structures: the Cu$_2$MnAl-type and the Hg$_2$CuTi-type. The former one is also known as L2$_1$-type structure (Fm-$\bar{3}$m space group, No. 225), where the two X atoms occupy A and C positions, and the latter one is known as XA-type structure (F-$\bar{4}$3m space group, No. 216), where the two X atoms occupy A and B positions. We select Fe$_2$NiSi as an example and show the two cubic crystal structures in figure 1*a,b*. For L2$_1$-type structure of Fe$_2$NiSi, the four Wyckoff sites A, B, C and D are occupied by Fe1, Ni, Fe2 and Si atoms, respectively. Whereas, for the XA-type structure, the four Wyckoff sites A, B, C and D are occupied by Fe1, Fe2, Ni and Si atoms, respectively.

In general, for full Heusler alloys, site preferences of transition metal elements X and Y are determined by the number of their valence electrons [75,76]: the element with more valence electrons prefers the A and C sites yet the element with less valence electrons prefers the B site. This rule has been widely used for explaining the atom ordering in Heusler alloy and even applied for new Heusler design. However, there are also some studies showing the contrary results which violate this rule [77,78]. It should be stressed that the atom ordering in Heusler alloys strongly influences their properties. In order to determine the stable state of cubic structure Fe$_2$NiSi and Ni$_2$FeSi, we computed their total energy with both L2$_1$-type and XA-type structures under different lattice constants. Besides, two magnetic states, ferromagnetic (FM) and non-magnetic (NM), have also been considered in each structure type. The results are shown in figure 2*a,b* for Fe$_2$NiSi and Ni$_2$FeSi, respectively.

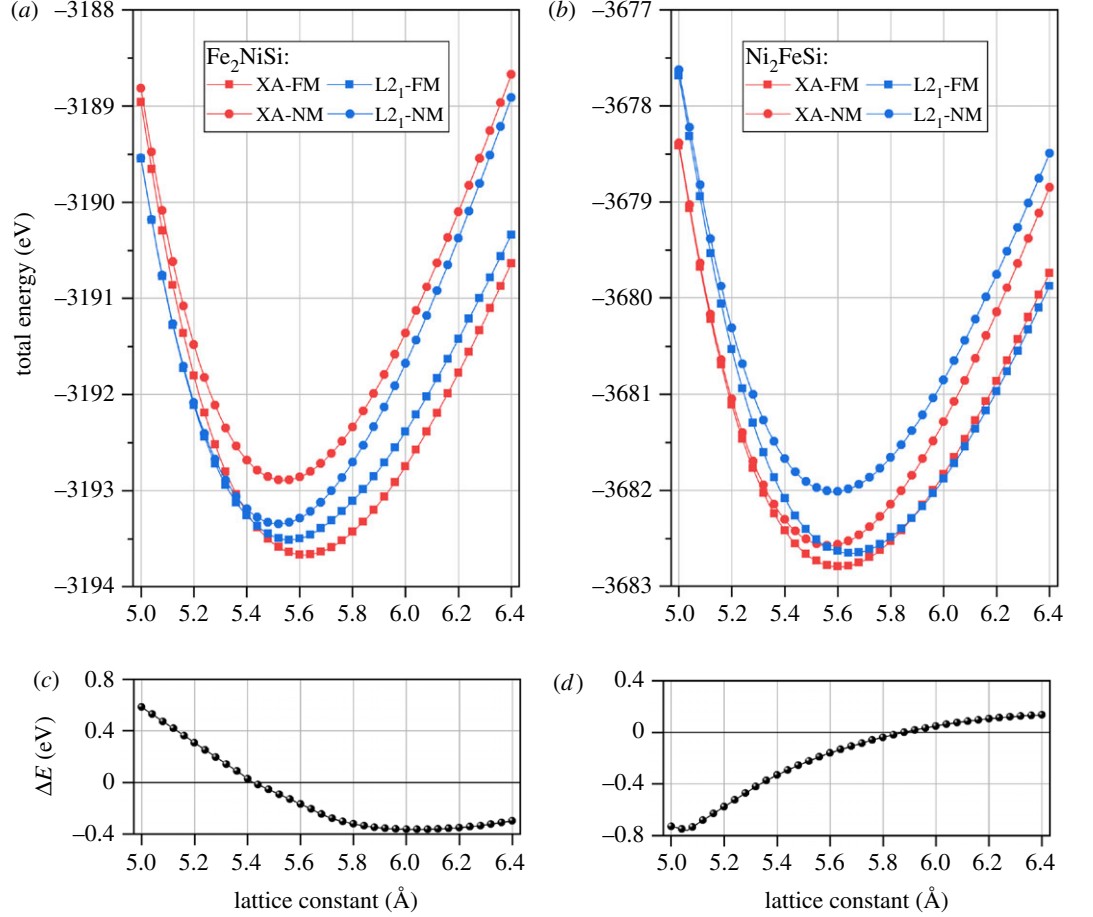

**Figure 2.** Total energy of full Heusler alloys $Fe_2NiSi$ (a) and $Ni_2FeSi$ (b) with different crystal structures under different lattice constants. The non-magnetic (NM) and ferromagnetic (FM) states are considered. (c,d) The total energy differences between ferromagnetic XA-type and ferromagnetic $L2_1$-type of $Fe_2NiSi$ and $Ni_2FeSi$, respectively.

It is seen from figure 2a,b that the non-magnetic states of both $L2_1$-type and XA-type structures have higher total energy than their ferromagnetic counterparts for both $Fe_2NiSi$ and $Ni_2FeSi$ Heusler alloys, meaning that the ferromagnetic state is more energetically stable. Moreover, the total energy curves of ferromagnetic states would converge to the non-magnetic ones at small lattice constants and this is due to the abatement of magnetic moment with decreasing lattice, which will be discussed later. The antiferromagnetic state is not considered simply because the experimentally synthesized $Fe_2NiSi$ compounds show ferromagnetic configuration with large magnetic moment [65].

More importantly, it is found the XA-type ferromagnetic structures have the lowest total energy for $Fe_2NiSi$ and $Ni_2FeSi$, which implies the XA-type structure is the stable state. For $Fe_2NiSi$, its XA structure obeys the general site preference rule: two Fe atoms have less valence electrons than Ni atom and they occupy the A and B sites. While for $Ni_2FeSi$, it does not follow this rule and two Ni atoms with more valence electrons enter the A and B sites forming the XA structure instead of $L2_1$ structure. In addition, some studies show that the pure metal structure of X element from the Heusler alloy $X_2YZ$ has influence on its atomic ordering [79,80], i.e. when X metal prefers a FCC or HCP structure, the $X_2YZ$ Heusler crystalizes in the $L2_1$ structure; while X metal with BCC structure would lead to the XA structure. This rule is also coincident with $Fe_2NiSi$, because pure Fe metal has BCC structure. But for $Ni_2FeSi$, pure Ni metal prefers FCC structure and this would predict the $L2_1$ structure, contrary to the stable XA structure. Since both $L2_1$ and XA structures have been experimentally demonstrated in Heusler alloys, we still keep these two structures for the following consideration.

The lattice constants are determined by minimizing the total energy for each state, and their values are listed in table 1. The results from our calculations are in good agreement with other experimental measurements and theoretical calculations [37,65–68]. In order to further check the structure stability, we computed the total energy difference between the XA structure and the $L2_1$ structure at ferromagnetic state under different lattice constants, and the results are plotted in figure 2c,d. For

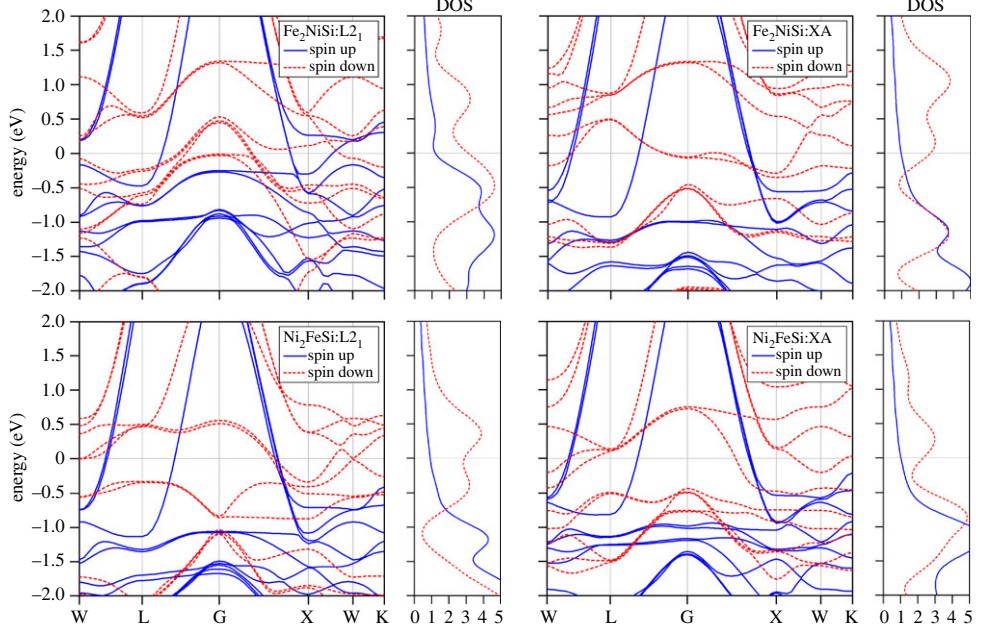

**Figure 3.** The calculated electronic band structure and density of states for $Fe_2NiSi$ and $Ni_2FeSi$ at the equilibrium lattice under two ferromagnetic crystal structures of $L2_1$-type and XA-type.

**Table 1.** The calculated equilibrium lattice constants and the corresponding magnetic moments of $Fe_2NiSi$ and $Ni_2FeSi$ in different crystal structures.

| compound | structure | lattice (Å) | | magnetic moment ($\mu_B$) | | | | |
| --- | --- | --- | --- | --- | --- | --- | --- | --- |
| | | present | other | total | A | B | C | D |
| $Fe_2NiSi$ | $L2_1$ | 5.56 | 5.58 [52] | 3.31 | 1.59 | 0.37 | 1.59 | −0.23 |
| | XA | 5.62 | 5.67 [65] | 4.73 | 1.80 | 2.72 | 0.28 | −0.07 |
| $Ni_2FeSi$ | $L2_1$ | 5.65 | | 3.07 | 0.09 | 2.93 | 0.09 | −0.04 |
| | XA | 5.61 | 5.61 [52] | 2.62 | 0.15 | 0.36 | 2.22 | −0.11 |

$Fe_2NiSi$, this energy difference is positive at small lattice constant, indicating the stability of $L2_1$ structure, and continuously decreases to negative value, leading to XA structure stabilization at large lattice constant. A reverse changing trend is observed for $Ni_2FeSi$ with stable XA structure at small lattice constant and stable $L2_1$ structure at large lattice constant. This sign changing effect of the total energy difference with lattice variation could induce the stable structure transformation under different lattice values. We also calculated the total energy difference of XA and $L2_1$ structures at ferromagnetic state under their own corresponding equilibrium lattice constants, and it is − 0.16 eV for $Fe_2NiSi$ and − 0.14 eV for $Ni_2FeSi$.

## 3.2. Electronic and magnetic properties

In this section, we calculate the electronic and magnetic properties of $Fe_2NiSi$ and $Ni_2FeSi$ compounds at the determined equilibrium lattice constants, via calculating the energy band structure, the densities of states, the electronic spin densities distribution and the magnetic moments. In order to check the atomic site preference in these two Heusler alloys and also investigate the reason for the contradictory behaviour observed for $Ni_2FeSi$, we include both $L2_1$ and XA structures. The electronic band structure and the corresponding density of state (DOS) for $Fe_2NiSi$ and $Ni_2FeSi$ in these two structures under their own equilibrium lattice are plotted in figure 3. It can be seen that the electronic bands exhibit an overlap with the Fermi energy level in both spin-up and spin-down directions for both $Fe_2NiSi$ and $Ni_2FeSi$ in either $L2_1$ or XA structures, which indicates their metallic nature. The effect of Hubbard U

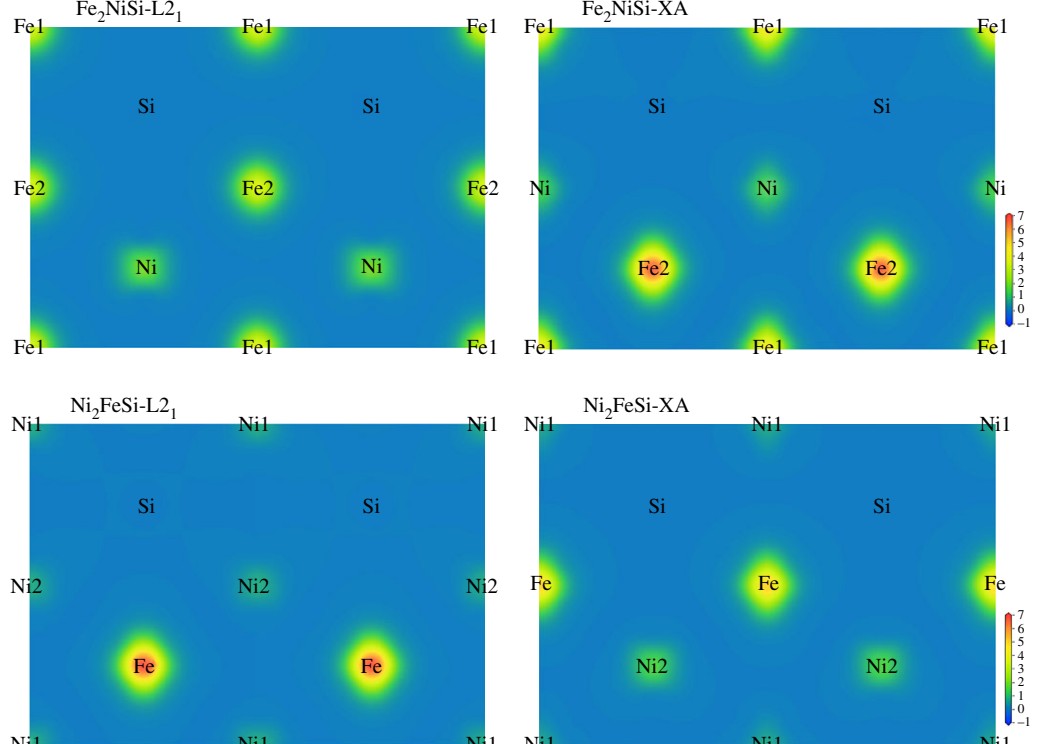

**Figure 4.** The calculated electronic spin density distributions in the (110) plane of Fe$_2$NiSi and Ni$_2$FeSi at the equilibrium lattice under two ferromagnetic crystal structures of L2$_1$-type and XA-type.

on the electronic band structure is also considered for the transition metal elements Fe and Ni, and results show that the metallic feature is still preserved even with some variation of the band structures. Usually, the DOS spectrum plays an important role for the structural stability of the intermetallic compounds [52]. In particular, a lower DOS at the Fermi energy level can induce a more stable structure. Thus, we compare the DOS of Fe$_2$NiSi and Ni$_2$FeSi at the Fermi energy level in L2$_1$ and XA structures. It is found that the DOS values of Fe$_2$NiSi in XA structure at the Fermi energy level is 0.96 states/eV and 2.65 states/eV in spin-up and spin-down channels, which are lower than 1.19 states/eV and 3.10 states/eV in L2$_1$ structure. For Ni$_2$FeSi, the DOS values at Fermi energy are 1.02 states/eV and 2.31 states/eV in XA structure and 0.91 states/eV and 2.90 states/eV for L2$_1$ structure in spin-up and spin-down channels, respectively, indicating the total DOS in XA structure is smaller than that in L2$_1$ structure. Consequently, we think the DOS values near the Fermi level have a dominant role of determining the stable XA structure in both Fe$_2$NiSi and Ni$_2$FeSi.

It is known that Fe-based ternary Heusler compounds often exhibit magnetism, which is very important for the application in spintronics and magnetoelectronics. So, we move on to examine the magnetic properties of Fe$_2$NiSi and Ni$_2$FeSi in different structures, and the calculated total and atom-resolved magnetic moments at equilibrium lattice are listed in table 1. The total magnetic moment is mainly contributed by the transition metal elements Fe and Ni, and also they are ferromagnetically aligned because of the same positive sign. It can also be found that the different atom site ordering leads to a larger variation in both total and atom-resolved magnetic moments.

To further elucidate the magnetic variation under different atomic orderings, we calculated the electronic spin density distribution of Fe$_2$NiSi and Ni$_2$FeSi in the (110) plane at the equilibrium lattice, and the results are displayed in figure 4. This electronic spin density is defined as the electronic density difference between the spin-up and spin-down directions and, thus, it can give information about the magnetic origination and distribution. Note that the same colourmap scale is applied for different structures so as to provide a better visual comparison. We can immediately see the bright colour around Fe atoms in all different structures, and this implies that Fe atoms have large spin density difference and strong magnetism. This can be simply understood because Fe atom often shows ferromagnetic properties in most Fe-based compounds. The spin density difference is smaller around Ni atoms and becomes even indistinguishable with background for Si atoms. For Fe$_2$NiSi in L2$_1$ structure, the colour around Fe atoms from two positions are all the same, leading to the same

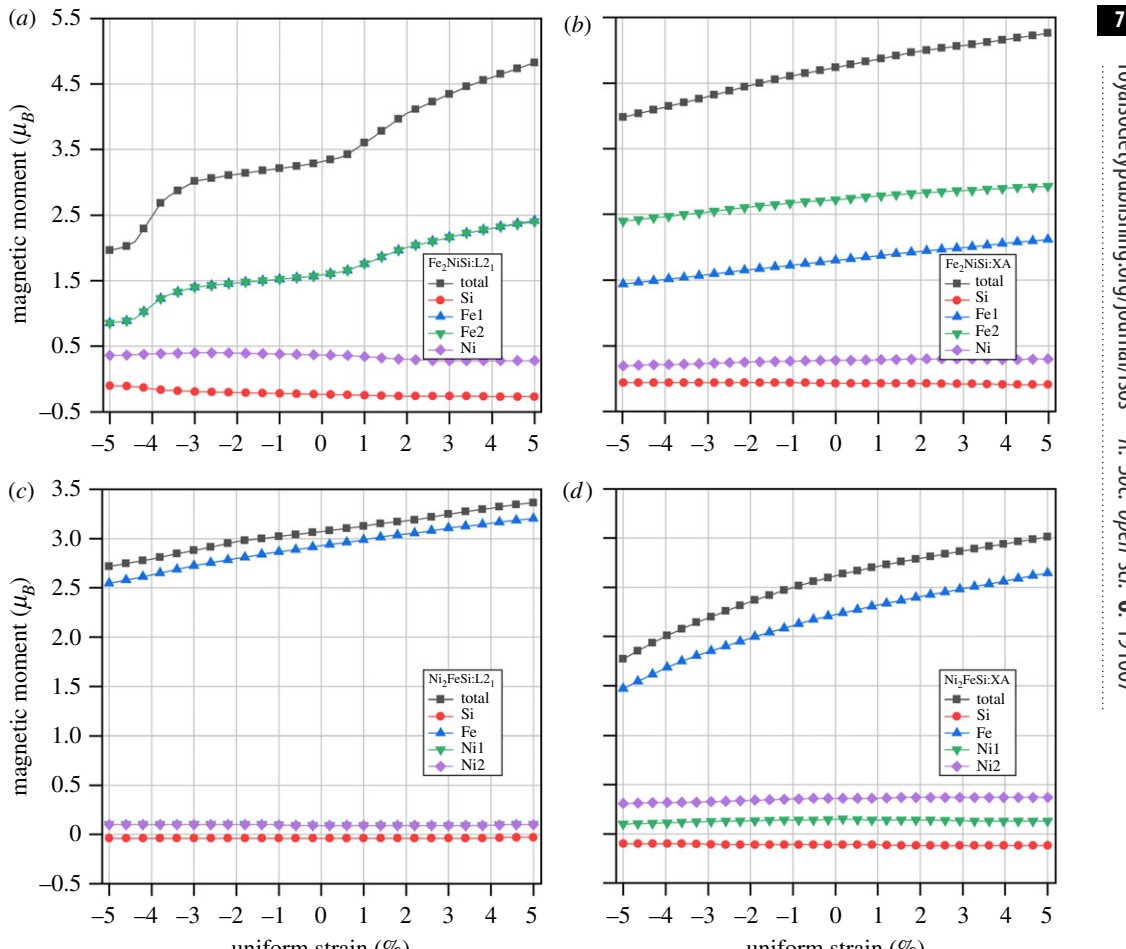

**Figure 5.** ($a$–$d$) Total and atomic spin magnetic moments of $Fe_2NiSi$ and $Ni_2FeSi$ under two ferromagnetic crystal structures of $L2_1$-type and XA-type as functions of the uniform strains. Atomic sites are referred to the crystal structure in figure 1.

magnetic moments as shown in table 1. This is because the two Fe atoms Fe1 and Fe2 from two sites have the same surrounding environment with four Ni atoms and four Si atoms as the nearest neighbours forming two tetrahedrons and six Fe atoms as the second nearest neighbours forming one octahedron. While, for the XA structure of $Fe_2NiSi$, the Fe atoms from two sites have different environments so they have different spin density distribution, as shown by a brighter colour and larger area around Fe2 than Fe1. Similar effects can also be found in $Ni_2FeSi$ with two Ni atoms of same environments and same magnetic moments under $L2_1$ structure yet different magnetic moments in XA structure. In order to investigate the atom site ordering and its impact on magnetic moments, we focus our inspection mainly on the Fe atoms. First, we compare the spin density distribution of $Fe_2NiSi$ in XA structure and $Ni_2FeSi$ in $L2_1$ structure with only one atom in A site changed from Fe to Ni. From table 1, we can see the magnetic moments of Fe atoms at B sites in these two compounds have very similar values, inferring that Ni atom as the nearest neighbour of Fe atom almost does not affect its magnetic moment when there is no Si atom present in the nearest neighbours of Fe atom. We suspect that the small difference could be introduced by the slightly different lattice constants. We compare the $Fe_2NiSi$ in $L2_1$ structure and $Ni_2FeSi$ in XA structure with only one atom in A site changed from Fe to Ni. The two Fe atoms in C sites both have much smaller magnetic moments than that of the two Fe atoms at B sites in the former two structures, which is probably caused by the different atomic configuration as Si atom is present in the nearest neighbours of Fe atom here and it suppresses the magnetism of Fe atom. Besides, the magnetic moments of Fe atoms, in this case, have larger difference and this could be still from the much bigger lattice difference. In combination, we found that Fe atom has much smaller magnetic moment when Si atom is present in its nearest neighbours and vice versa in both $Fe_2NiSi$ and $Ni_2FeSi$.

Afterwards, we also study the effect of uniform strain on the magnetic moments by varying the lattice constant around the equilibrium condition, and the results are reported in figure 5. It can be seen that the

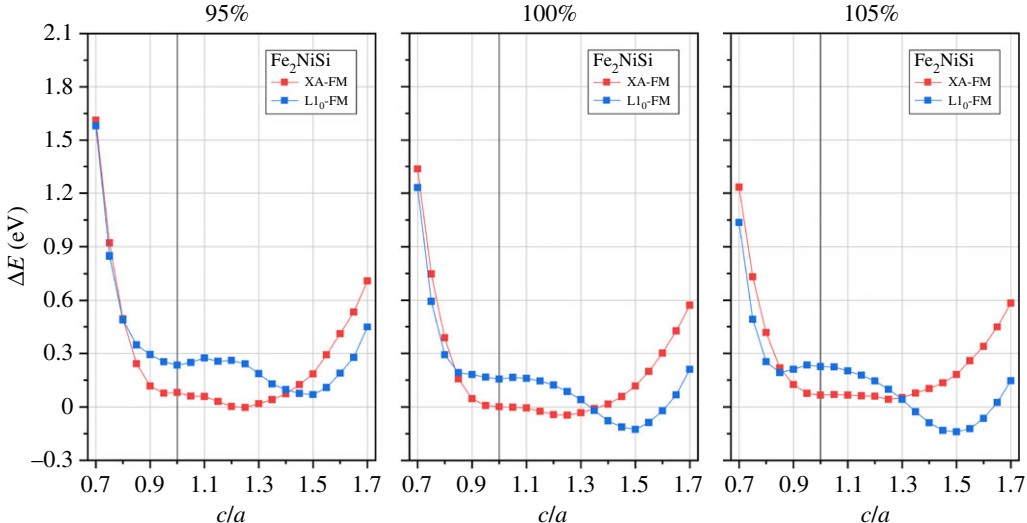

**Figure 6.** Total energy differences of $Fe_2NiSi$ in tetragonal $L1_0$ and XA structures under ferromagnetic state as functions of $c/a$ ratio. Three percentage values of the corresponding cubic structure equilibrium volume are considered and indicated at each panel top. The zero energy point is set as the XA-FM structure of $c/a$ equal to 1 at 100% volume.

total and atom-resolved magnetic moments of $Fe_2NiSi$ and $Ni_2FeSi$ in both $L2_1$ and XA structures increase with lattice expanded at positive strain and decrease with lattice contracted at negative strain. For the $L2_1$ structure in both $Fe_2NiSi$ and $Ni_2FeSi$, the magnetic moments in A and C sites always overlap with each other (Fe1 and Fe2, Ni1 and Ni2) throughout the whole strain variation because they have the same surrounding environment as explained above. The variation of the total magnetic moment is mainly from the Fe atoms and there is almost no change for Si atoms for all cases. While for Ni atoms, different changing trends are observed: a small decrease in $L2_1$ structure and a slightly bigger increase in XA structure for $Fe_2NiSi$; no changes in $L2_1$ structure and a small increase in XA structure for $Ni_2FeSi$. With lattice decrease, the distance between atoms gets smaller so that the interaction of the valence electrons from different atoms is enhanced and they become more delocalized. The larger variation with strain found from $Fe_2NiSi$ in $L2_1$ structure and $Ni_2FeSi$ in XA structure is all related with Fe atom and this is because of the surrounding Si atom as the nearest neighbours of Fe atom. With strain increase at positive side, the distance between Fe and Si atoms is larger so that Fe atom partially regain its spin moments, and vice versa. In particular, for $Fe_2NiSi$ in XA structure, the magnetic moment variation of Fe1 is a little larger than Fe2 and this is probably caused by the different neighbouring distance of Si atoms: Si atom is nearest to Fe1 while it is secondary nearest to Fe2. The electronic band structures for both $Fe_2NiSi$ and $Ni_2FeSi$ under uniform strain have also been calculated and they all show metallic overlap with the Fermi energy in both spin-up and spin-down directions, indicating the metallic behaviour under studied strain condition.

## 3.3. Tetragonal transformation

According to the theoretical calculation and experimental study [52,69], both Fe-based and Ni-based Heusler compounds may undergo a tetragonal distortion and transform from cubic phase into tetragonal phase. This phase transformation is still related with the crystal structure arrangement and, thus, we further investigate the tetragonal distortion in terms of $c/a$ ratio variation in both $L2_1$ and XA structures under ferromagnetic state. Note, when the $c/a$ ratio is varied, the unit cell volume is fixed at the equilibrium condition. The corresponding structures for $Fe_2NiSi$ in tetragonal $L1_0$-type and tetragonal XA-type are shown in figure 1c,d, respectively. In order to understand the phase stability, we first calculate the total energy at different $c/a$ ratios, and the results are presented in figures 6 and 7 for $Fe_2NiSi$ and $Ni_2FeSi$, respectively.

It is clearly seen that the tetragonal distortion can reduce the total energy for both $Fe_2NiSi$ and $Ni_2FeSi$ in either $L2_1$ or XA structure, which may lead to the phase transformation. As experimentally proved, $Fe_2NiSi$ prepared by arc-melting crystallizes in cubic structure [66] and this is due to the $L2_1B$ disorder effect. Also, obtained elastic modulus constants from previous study [68] confirmed the mechanical stability for $Fe_2NiSi$ in cubic structure. Thus, the possible tetragonal phase transformation

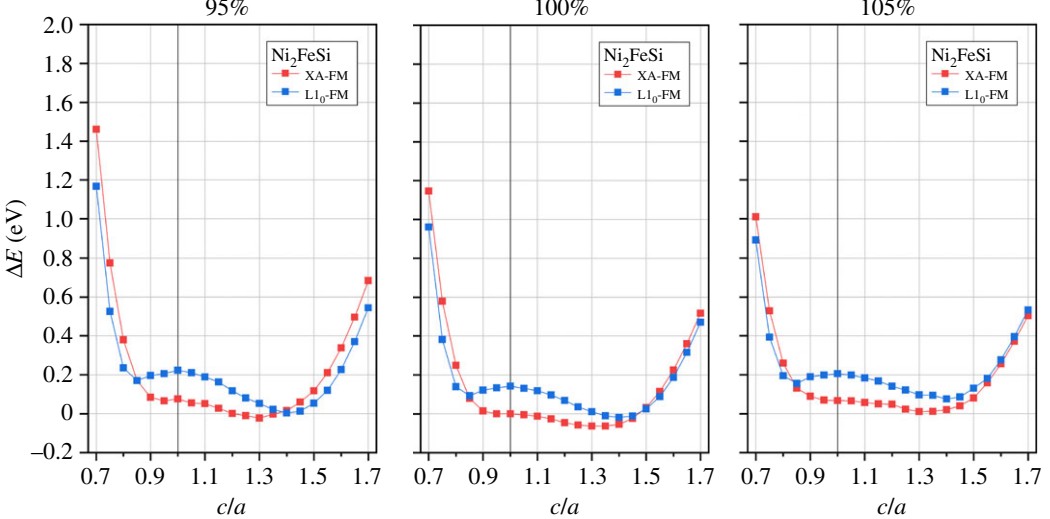

**Figure 7.** Total energy differences of $Ni_2FeSi$ in tetragonal $L1_0$ and XA structures under ferromagnetic state as functions of $c/a$ ratio. Three percentage values of the corresponding cubic structure equilibrium volume are considered and indicated at each panel top. The zero energy point is set as the XA-FM structure of $c/a$ equal to 1 at 100% volume.

is not caused by mechanical properties. Another comprehensive theoretical calculation from the energetic point of view shows that $Fe_2NiSi$ has tetragonal $L1_0$ ground state and attributes the reason to the density of states at the Fermi energy level [52]. This transformation is very interesting because not only the tetragonal structural distortion but also the atom site reordering occurs. From figure 6 at 100% volume, we can see that $Fe_2NiSi$ at cubic condition ($c/a = 1$) has stable XA structure because of the low total energy, as discussed before, and, with the variation of $c/a$ value, its total energy decreases much more strongly in tetragonal $L1_0$ structure than in tetragonal XA structure to the minimum total energy when $c/a$ ratio is equal to 1.5, meaning the stable cubic XA structure could change into tetragonal $L1_0$ structure with tetragonal distortion. This finding is consistent with previous study [52]. By varying the unit cell volume to 95% and 105%, an opposite effect is observed for the stable tetragonal distortion: the tetragonal XA structure has the lowest total energy at 95% volume, meaning that only tetragonal structural distortion is present without atom site reordering; while for 105% volume, the atom site ordering and tetragonal structural distortion have been enhanced because of the larger total energy difference.

For $Ni_2FeSi$ as shown in figure 7, the tetragonal distortion can still decrease the total energy and may lead to the phase transformation. Through the 95% to 105% volume variation, the tetragonal XA structure always has the minimum energy value, indicating the presence of only the tetragonal structural distortion and no atom site reordering occurs, which is different from the case of $Fe_2NiSi$. However, by observing the energy difference trend at different volume percentages, the minimum total energies under tetragonal $L1_0$ and XA structures are getting close to each other at smaller volume, meaning the similar stability of these two structures. In combination, it is found the tetragonal structural distortion can decrease the total energy of cubic $Fe_2NiSi$ and $Ni_2FeSi$ in either $L2_1$ or XA structure and possibly lead to phase transformation. The origin of these tetragonal ground states of $Fe_2NiSi$ and $Ni_2FeSi$ has been theoretically explained by the decreasing effect of the density of states at the Fermi energy level [52]. At the same time, atom site reordering may occur in this process dependent on the cell volume variation.

In addition, to further evaluate the stability of the tetragonal phase, we define the tetragonal transformation energy ($\Delta E_T$) as the difference of the total energy at cubic equilibrium state and the minimum total energy at tetragonal distortion. Its variation under different volume percentages for both $Fe_2NiSi$ and $Ni_2FeSi$ is displayed in figure 8. The positive sign of all the values shows the stable tetragonal phase under currently studied volume range. With volume increase, the $\Delta E_T$ values all decrease except $Fe_2NiSi$ in tetragonal $L1_0$ structure, meaning the structural stability of the tetragonal phase is becoming weaker at larger volume. It is also found that the values of $\Delta E_T$ in tetragonal XA structure are much smaller than in tetragonal $L1_0$ structure, especially at larger volume side, which indicates the stronger stability in tetragonal $L1_0$ phase. It should be pointed out that the values of $c/a$ ratio at the minimum total energy under tetragonal phase for $Fe_2NiSi$ and $Ni_2FeSi$ in different

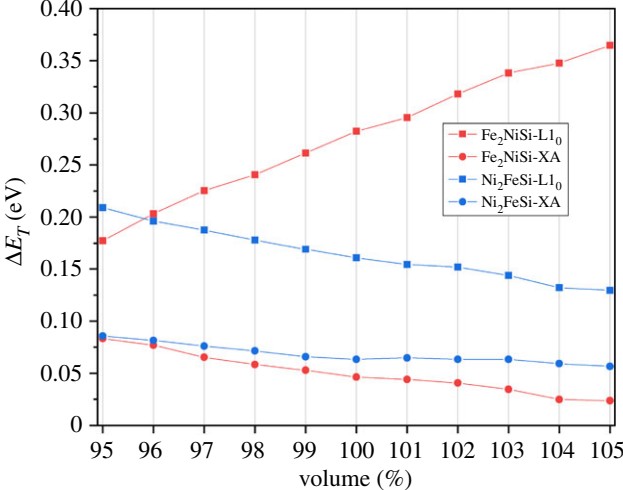

**Figure 8.** The tetragonal transformation energy ($\Delta E_T$) under different volume percentages.

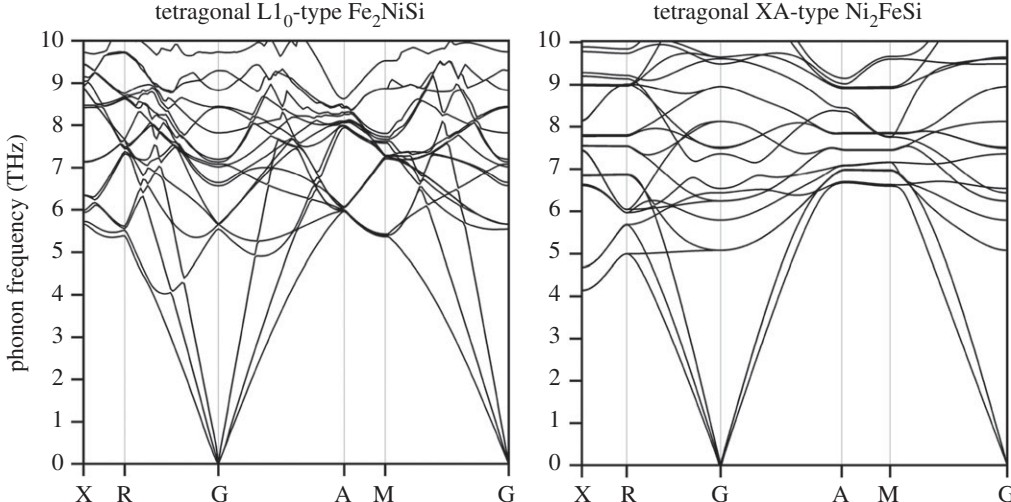

**Figure 9.** The calculated photon dispersion spectra of Fe$_2$NiSi and Ni$_2$FeSi in tetragonal phases.

structures remain almost unchanged with volume variation: 1.25 and 1.50 for Fe$_2$NiSi and 1.30 and 1.40 for Ni$_2$FeSi in tetragonal XA and L1$_0$ structures, respectively. In order to check the dynamic stability of the tetragonal phase, the phonon dispersion spectra for Fe$_2$NiSi and Ni$_2$FeSi are also calculated and are displayed in figure 9. It is clearly seen that the phonon curve in the tetragonal phases for the two compounds exhibits no imaginary frequency and, therefore, it indicates the dynamic stability of the tetragonally distorted structure. Besides, the electronic band structures calculated at the minimum total energy point of tetragonal distortion for both Fe$_2$NiSi and Ni$_2$FeSi have been calculated, and they all have metallic overlap with the Fermi energy in both spin directions.

As shown in §3.1 and also demonstrated in both experiment and theory, cubic structure Fe$_2$NiSi and Ni$_2$FeSi Heusler compounds are ferromagnets with considerably large magnetic moments. Magnetic variation under tetragonal distortion is also of high interest and great importance because it is directly related with the magnetic shape memory effect and the development of spin-transfer torque magnetic random access memory. The calculated total and atom-resolved magnetic moments for Fe$_2$NiSi and Ni$_2$FeSi in both tetragonal L1$_0$ and tetragonal XA structures under different $c/a$ ratios are shown in figures 10 and 11, respectively. We can see that the variation of the total magnetic moment is still mainly from the Fe atoms, the same as in the cubic structure. Consequently, we focus our discussion on the magnetic moment of Fe atoms. For the tetragonal L1$_0$ structure in both Fe$_2$NiSi and Ni$_2$FeSi, the two atoms in sites A and C always have the same moment values and thus overlap throughout the whole $c/a$ variation because of the same surrounding environment even in tetragonal structure.

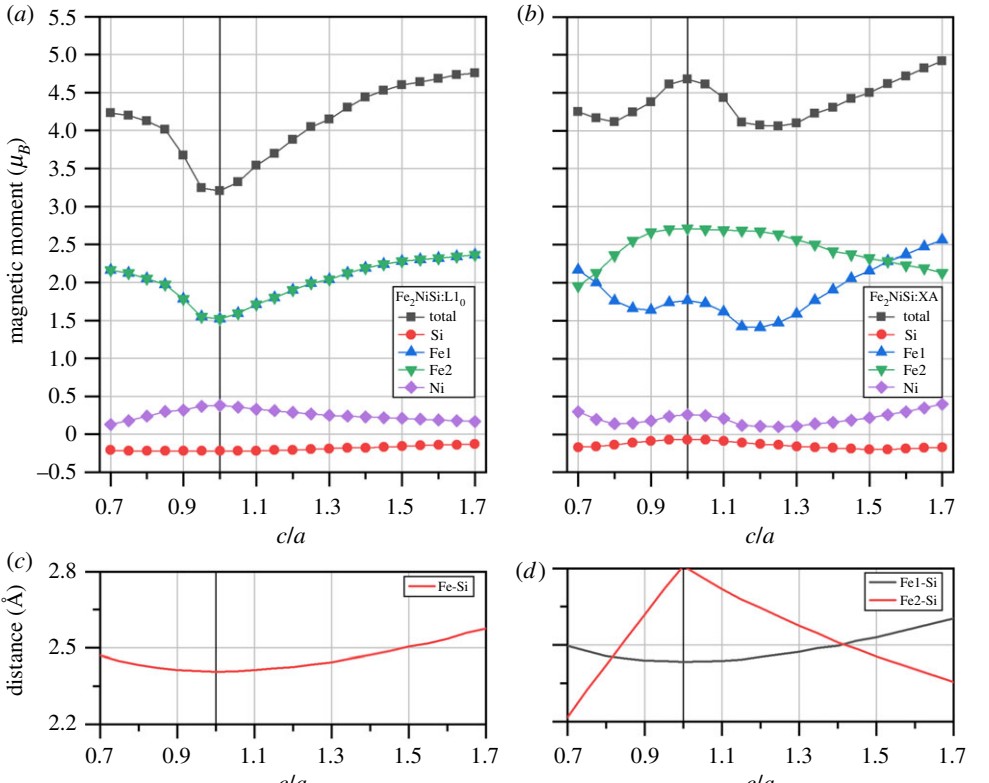

**Figure 10.** The calculated total and atomic spin magnetic moments of Fe$_2$NiSi under two ferromagnetic crystal structures of L1$_0$-type (*a*) and XA-type (*b*) as functions of *c/a* ratio. Atomic sites are referred to the crystal structure in figure 1. The distance between Fe atom and its nearest Si atom in the two corresponding structures are shown in (*c,d*).

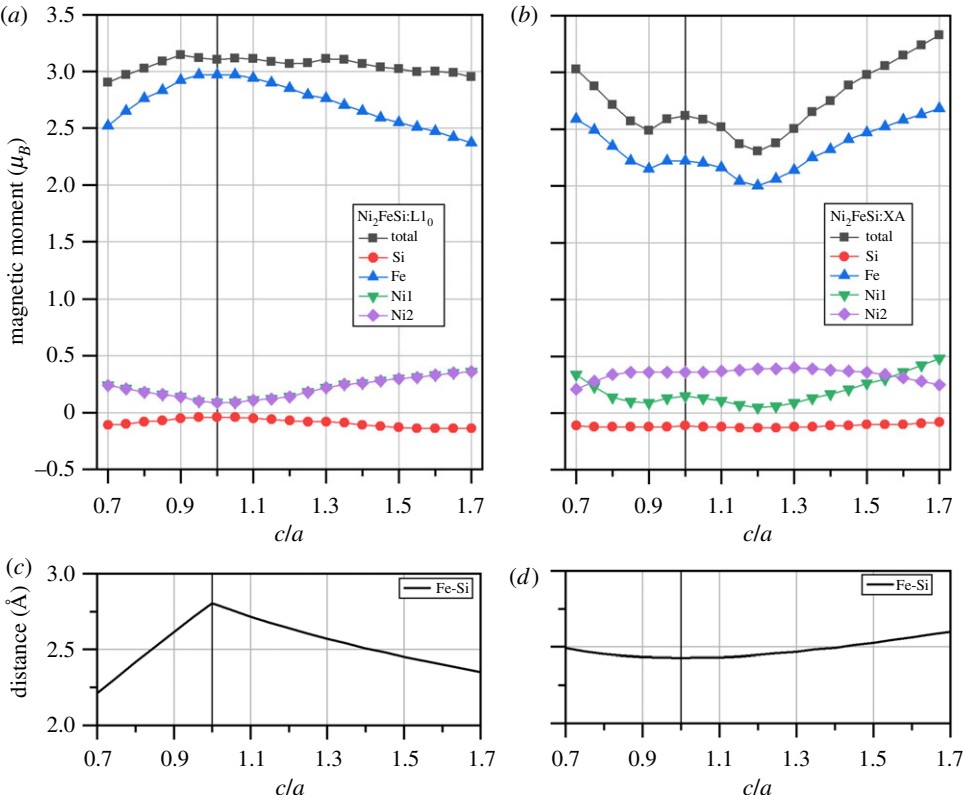

**Figure 11.** The calculated total and atomic spin magnetic moments of Ni$_2$FeSi under two ferromagnetic crystal structures of L1$_0$-type (*a*) and XA-type (*b*) as functions of *c/a* ratio. Atomic sites are referred to the crystal structure in figure 1. The distance between Fe atom and its nearest Si atom in the two corresponding structures are shown in (*c,d*).

In order to detail the moment variation of Fe atom, we need to go back to the crystal structure. As an example for $Fe_2NiSi$ in tetragonal $L1_0$ structure, the two Fe atoms in sites A and C have the same surrounding atoms of four Si atoms and four Ni atoms forming two tetrahedrons, see figure 1$c$. By varying the $c/a$ ratio, the distance between Fe atom and its nearest Si atom is calculated and shown in figure 10$c$. It is found that this distance value is increased when the $c/a$ ratio is changed from 1 on both sides. As discussed in §3.2, this distance plays an important role for the magnetic moment of Fe atom and its increase would lead to the moment gain of Fe atom, as found in figure 1$a$. For $Fe_2NiSi$ in tetragonal XA structure, the two Fe atoms have different environments so that we need to calculate the nearest Fe–Si distance for each of them, and the results are shown in figure 10$d$. The magnetic moment variation of the two atoms can be still related with the Fe–Si distance change in the same way. For $Ni_2FeSi$ in both tetragonal $L1_0$ and tetragonal XA structures, the partial moment variation of Fe atom can still be understood with the Fe–Si distance in the same way.

## 4. Conclusion

In the current work, we systematically studied the structural configuration of the full Heusler compounds $Fe_2NiSi$ and $Ni_2FeSi$ by employing the first-principles calculations based on density functional theory, in terms of the structural, electronic and magnetic properties. Besides, the effects of uniform and tetragonal strains have been also considered and discussed. Results show that both $Fe_2NiSi$ and $Ni_2FeSi$ prefer XA-type atomic ordering rather than $L2_1$-type in cubic phase due to the lower total energy. The obtained lattice constants agree with previous experimental and theoretical studies. Both $Fe_2NiSi$ and $Ni_2FeSi$ show metallic band structures and large magnetic moments (greater than $3\mu_B$) at equilibrium condition. Under tetragonal distortion, total energy can be further decreased, leading to the possible phase transformation, but different atom site reordering behaviours have been observed: for $Fe_2NiSi$, tetragonal $L1_0$ structure has smaller total energy than tetragonal XA structure at 100% unit cell volume, which implies that atoms reorder from cubic XA-type to tetragonal $L1_0$-type; for $Ni_2FeSi$, the XA structure always has the lowest total energy, which means there is only structural transformation without atom reordering. This atom reordering behaviour under tetragonal phase transformation is very interesting and can open up a variation of studies for material property under distortion conditions. The total magnetic moments of $Fe_2NiSi$ and $Ni_2FeSi$ are mainly contributed by Fe atoms, and Si atom can strongly suppress the moments of Fe atoms when Si atom is present in the nearest neighbours of Fe atoms. With strain applied, the distance between Fe and Si atoms plays an important role for the magnetic moment variation of Fe atom. Moreover, the metallic band structure is maintained for $Fe_2NiSi$ and $Ni_2FeSi$ under both uniform and tetragonal strains. Overall, this study provides a detailed theoretical analysis and can give valuable reference for further experimental research.

Data accessibility. The datasets supporting this article have been uploaded as part of the electronic supplementary material.

Authors' contributions. L.H. and T.Y. conceived and performed the theoretical calculations. R.K. and X.W. constructed and revised the manuscript. All authors gave final approval for publication.

Competing interests. We declare we have no competing interests.

Funding. We received no funding for this study.

Acknowledgements. The authors thank the anonymous reviewers for their constructive suggestions.

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
