## [Reviewer comments · Royal Society Open Science]

Review History

RSOS-191007.R0 (Original submission)

Review form: Reviewer 1

Is the manuscript scientifically sound in its present form?

Yes

Are the interpretations and conclusions justified by the results?

No

Is the language acceptable?

Yes

Do you have any ethical concerns with this paper?

No

Have you any concerns about statistical analyses in this paper?

No

Recommendation?

Major revision is needed (please make suggestions in comments)

Comments to the Author(s)

In this paper, the authors have investigated the structural, electronic, and magnetic properties of Heusler Fe₂NiSi and Ni₂FeSi compounds. The detailed strain effect is also reported in the paper. However, I think the authors should perform more calculations to check the obtained result:

1. For the 3d Fe and Ni systems, the Hubbard U should be considered in the calculations. The authors should do a GGA+U calculation to check the obtained results.
2. Do the authors consider the possible AFM state in these Fe₂NiSi and Ni₂FeSi systems? The authors should confirm the FM state is more stable than the AFM one.
3. Are the investigated systems mechanically stable? Is the tetragonal transformation related to the mechanical instability of cubic phase?

Review form: Reviewer 2

Is the manuscript scientifically sound in its present form?

Yes

Are the interpretations and conclusions justified by the results?

Yes

Is the language acceptable?

Yes

Do you have any ethical concerns with this paper?

No

Have you any concerns about statistical analyses in this paper?

No

Recommendation?

Accept with minor revision (please list in comments)

Comments to the Author(s)

Attending to the minor observations about the conclusions, I believe that the work should be accepted for publication.

Decision letter (RSOS-191007.R0)

08-Jul-2019

Dear Professor Khenata:

Title: Investigation of the structural competing and atomic ordering in Heusler compounds Fe₂NiSi and Ni₂FeSi under strain condition

Manuscript ID: RSOS-191007

The editor assigned to your manuscript has now received comments from reviewers. We would like you to revise your paper in accordance with the referee and Subject Editor suggestions which can be found below (not including confidential reports to the Editor). Please note this decision does not guarantee eventual acceptance.

Please submit your revised paper before 31-Jul-2019. Please note that the revision deadline will expire at 00.00am on this date. If we do not hear from you within this time then it will be assumed that the paper has been withdrawn. In exceptional circumstances, extensions may be possible if agreed with the Editorial Office in advance. We do not allow multiple rounds of revision so we urge you to make every effort to fully address all of the comments at this stage. If deemed necessary by the Editors, your manuscript will be sent back to one or more of the original reviewers for assessment. If the original reviewers are not available we may invite new reviewers.

Please also include the following statements alongside the other end statements. As we cannot publish your manuscript without these end statements included, if you feel that a given heading is not relevant to your paper, please nevertheless include the heading and explicitly state that it is not relevant to your work.

- Ethics statement

Please clarify whether you received ethical approval from a local ethics committee to carry out your study. If so please include details of this, including the name of the committee that gave consent in a Research Ethics section after your main text. Please also clarify whether you received informed consent for the participants to participate in the study and state this in your Research Ethics section.

OR

Please clarify whether you obtained the necessary licences and approvals from your institutional animal ethics committee before conducting your research. Please provide details of these licences and approvals in an Animal Ethics section after your main text.

OR

Please clarify whether you obtained the appropriate permissions and licences to conduct the fieldwork detailed in your study. Please provide details of these in your methods section.

- Data accessibility

It is a condition of publication that you make available the data and research materials supporting the results in the article. Datasets should be deposited in an appropriate publicly available repository and details of the associated accession number, link or DOI to the datasets must be included in the Data Accessibility section of the article

(<http://royalsocietypublishing.org/instructions-authors#question17>). Reference(s) to datasets should also be included in the reference list of the article with DOIs (where available).

Please include a Data Availability section after your main text stating where supporting data are available from, or where they will be made available should your article be accepted for publication.

If you wish to submit your supporting data or code to Dryad (<http://datadryad.org/>), or modify your current submission to dryad, please use the following link:
<http://datadryad.org/submit?journalID=RSOS&manu=RSOS-191007>

- **Competing interests**

Please include a Competing Interests section after your main text declaring any financial or non-financial competing interests. If you have no competing interests please state 'I/we have no competing interests.'

- **Authors' contributions**

Please include an Authors' Contributions section at the end of your main text detailing the contribution of each author. All authors should have read and approved the manuscript before submission and this should be stated in the Authors' Contributions section.

The list of Authors should meet all of the following criteria; 1) substantial contributions to conception and design, or acquisition of data, or analysis and interpretation of data; 2) drafting the article or revising it critically for important intellectual content; and 3) final approval of the version to be published.

- **Acknowledgements**

- **Funding statement**

Please include a funding section after your main text which lists the source of funding for each author.

RSC Associate Editor:

Comments to the Author:

Reviewer 2 commented that the conclusions do not mention the partial results on the XA and L21 structures. Please include these results in the conclusions.

RSC Subject Editor:

Comments to the Author:

(There are no comments.)

Reviewers' Comments to Author:

Reviewer: 1

Comments to the Author(s)

In this paper, the authors have investigated the structural, electronic, and magnetic properties of Heusler Fe₂NiSi and Ni₂FeSi compounds. The detailed strain effect is also reported in the paper. However, I think the authors should perform more calculations to check the obtained result:

1. For the 3d Fe and Ni systems, the Hubbard U should be considered in the calculations. The authors should do a GGA+U calculation to check the obtained results.
2. Do the authors consider the possible AFM state in these Fe₂NiSi and Ni₂FeSi systems? The authors should confirm the FM state is more stable than the AFM one.
3. Are the investigated systems mechanically stable? Is the tetragonal transformation related to the mechanical instability of cubic phase?

Reviewer: 2

Comments to the Author(s)

Attending to the minor observations about the conclusions, I believe that the work should be accepted for publication.

Author's Response to Decision Letter for (RSOS-191007.R0)

See Appendix A.

RSOS-191007.R1 (Revision)

Review form: Reviewer 1

Is the manuscript scientifically sound in its present form?

Yes

Are the interpretations and conclusions justified by the results?

Yes

Is the language acceptable?

Yes

Do you have any ethical concerns with this paper?

No

Have you any concerns about statistical analyses in this paper?

Yes

Recommendation?

Accept as is

Comments to the Author(s)

The revised paper is suitable for the RSOS.

Decision letter (RSOS-191007.R1)

12-Aug-2019

Dear Professor Khenata:

Title: Investigation of the structural competing and atomic ordering in Heusler compounds

Fe_2NiSi and Ni_2FeSi under strain condition

Manuscript ID: RSOS-191007.R1

It is a pleasure to accept your manuscript in its current form for publication in Royal Society Open Science. The chemistry content of Royal Society Open Science is published in collaboration with the Royal Society of Chemistry.

RSC Associate Editor:
Comments to the Author:
(There are no comments.)

RSC Subject Editor:
Comments to the Author:
(There are no comments.)

Reviewer(s)' Comments to Author:
Reviewer: 1

Comments to the Author(s)
The revised paper is suitable for the RSOS.

Appendix A

RESPONSE

We would like to thank the editor and reviewer for processing our paper and providing very useful comments. We carefully read the reports and revised the manuscript according to them. All the changes to the original manuscript are shown in red-colored font. Detailed answer for editor and reviewer is listed below.

RSC Associate Editor:

Comments to the Author:

Reviewer 2 commented that the conclusions do not mention the partial results on the XA and L21 structures. Please include these results in the conclusions.

Answer: The corresponding part on the XA and L21 structures have been added in the conclusions part.

Reviewers' Comments to Author:

Reviewer: 1

Comments to the Author(s)

In this paper, the authors have investigated the structural, electronic, and magnetic properties of Heusler Fe_2NiSi and Ni_2FeSi compounds. The detailed strain effect is also reported in the paper. However, I think the authors should perform more calculations to check the obtained result:

1. For the 3d Fe and Ni systems, the Hubbard U should be considered in the calculations.

The authors should do a GGA+U calculation to check the obtained results.

Answer: The reviewer is correct about the Hubbard U for transition metals elements Fe and Ni. However, both experimental and theoretical studies from Luo et al. [*Journal of Magnetism and Magnetic Materials*, 419, 485 (2016); *Journal of Physics D: Applied Physics*, 40, 7121 (2007)] have been done in Fe₂NiSi and the application of GGA in the calculations has been confirmed to be consistent with both experimental results and other theoretical methods, such as FLAPW and CPA. In order to further evaluate the effect of Hubbard U, we calculate the band structure of Fe₂NiSi and Ni₂FeSi in XA-type cubic structure, see the following figures, and it can be found the even the band structures have variation with U applied, the metallic nature is preserved for both compounds. Thus, we think the GGA methods applied in current work is sufficient to support the results obtained. Some text has been revised to address this point in the manuscript.

Fig1. Electronic band structures for Fe₂NiSi in XA-type cubic structure with GGA a) and GGA+U b) methods.

Fig2. Electronic band structures for Ni₂FeSi in XA-type cubic structure with GGA a) and GGA+U b) methods.

2. Do the authors consider the possible AFM state in these Fe₂NiSi and Ni₂FeSi systems?

The authors should confirm the FM state is more stable than the AFM one.

Answer: The FM state in Fe₂NiSi system has been investigated both theoretically and experimentally. Such as, Zhang et al. [*Physica B*, 420, 86 (2013)] determined the total magnetic moment of Fe₂NiSi is 4.10 μ_B from experiment measurements and this value matches our calculation result considerably well. For Ni₂FeSi system, it can be considered as Ni-rich doped system from Fe₂NiSi. Also, another theoretical calculation study [*Physical Review Applied*, 7, 034022 (2017)] considered the FM state as the ground state. Thus, we believe the FM state should be more stable than the AFM one and we only adopt the FM state in this work. In accordance, some extra explanation is added in Section 3 (a) to clarify this point.

3. Are the investigated systems mechanically stable? Is the tetragonal transformation related to the mechanical instability of cubic phase?

Answer: The reviewer is very considerate to raising this point about the mechanical stability of the investigated systems. Indeed, Gupta et al. [*Materials Chemistry and Physics*, 146, 303 (2014)] have already calculated the elastic modulus of Fe₂NiSi in cubic XA structure: C₁₁=219.63 GPa, C₁₂=153.34 GPa and C₄₄=143.63 GPa. From their obtained results, it can be derived on the basis of the general elastic stability criteria that cubic Fe₂NiSi is mechanically stable. Thus, the tetragonal transformation is not really related with the mechanical stability. Faleev et al. [*Physical Review Applied*, 7, 034022 (2017)] have carried a systematic and comprehensive study about the origin of the tetragonal ground state in 286 Heusler compounds, including the currently studied two compounds Fe₂NiSi and Ni₂FeSi, and attributed it to the peak-and-valley character of the density of states, which can be shifted to lower value by tetragonal transformation. This situation has already been mentioned in the manuscript and some more sentences are further inserted to stress it more clearly.

Reviewer: 2

Comments to the Author(s)

Attending to the minor observations about the conclusions, I believe that the work should be accepted for publication.

Answer: The corresponding part on the XA and L21 structures have been added in the conclusions part.